# Subspace Clustering with Irrelevant Features via Robust Dantzig Selector

**Chao Qu**
Department of Mechanical Engineering
National University of Singapore

A0117143@u.nus.edu

**Huan Xu**
Department of Mechanical Engineering
National University of Singapore

mpexuh@nus.edu.sg

## Abstract

This paper considers the subspace clustering problem where the data contains irrelevant or corrupted features. We propose a method termed "robust Dantzig selector" which can successfully identify the clustering structure even with the presence of irrelevant features. The idea is simple yet powerful: we replace the inner product by its robust counterpart, which is insensitive to the irrelevant features given an upper bound of the number of irrelevant features. We establish theoretical guarantees for the algorithm to identify the correct subspace, and demonstrate the effectiveness of the algorithm via numerical simulations. To the best of our knowledge, this is the first method developed to tackle subspace clustering with irrelevant features.

## 1 Introduction

The last decade has witnessed fast growing attention in research of high-dimensional data: images, videos, DNA microarray data and data from many other applications all have the property that the dimensionality can be comparable or even much larger than the number of samples. While this setup appears ill-posed in the first sight, the inference and recovery is possible by exploiting the fact that high-dimensional data often possess low dimensional structures [3, 14, 19]. On the other hand, in this era of big data, huge amounts of data are collected everywhere, and such data is generally heterogeneous. Clean data and irrelevant or even corrupted information are often mixed together, which motivates us to consider the high-dimensional, big but dirty data problem. In particular, we study the subspace clustering problem in this setting.

Subspace clustering is an important subject in analyzing high-dimensional data, inspired by many real applications[15]. Given data points lying in the union of multiple linear spaces, subspace clustering aims to identify all these linear spaces, and cluster the sample points according to the linear spaces they belong to. Here, different subspaces may correspond to motion of different objects in video sequence [11, 17, 20], different rotations, translations and thickness in handwritten digit or the latent communities for the social graph [15, 5].

A variety of algorithms of subspace clustering have been proposed in the last several years including algebraic algorithms [16], iterative methods [9, 1], statistical methods [11, 10], and spectral clustering-based methods [6, 7]. Among them, sparse subspace clustering (SSC) not only achieves state-of-art empirical performance, but also possesses elegant theoretical guarantees. In [12], the authors provide a geometric analysis of SSC which explains rigorously why SSC is successful even when the subspaces are overlapping [12]. [18] and [13] extend SSC to the noisy case, where data are contaminated by additive Gaussian noise. Different from these work, we focus on the case where some irrelevant features are involved.

Mathematically, SSC indeed solves for each sample a sparse linear regression problem with the dictionary being all other samples. Many properties of sparse linear regression problem are well understood in the clean data case. However, the performance of most standard algorithms deteriorates (e.g. LASSO and OMP) even only a few entries are corrupted. As such, it is well expected that standard SSC breaks for subspace clustering with irrelevant or corrupted features (see Section 5 for numerical evidences). Sparse regression under corruption is a hard problem, and few work has addressed this problem [8][21] [4].

**Our contribution:** Inspired by [4], we use a simple yet powerful tool called robust inner product and propose the robust Dantzig selector to solve the subspace clustering problem with irrelevant features. While our work is based upon the robust inner product developed to solve robust sparse regression, the analysis is quite different from the regression case since both the data structures and the tasks are completely different: for example, the RIP condition – essential for sparse regression – is hardly satisfied for subspace clustering [18]. We provide sufficient conditions to ensure that the Robust Dantzig selector can detect the true subspace clustering. We further demonstrate via numerical simulation the effectiveness of the proposed method. To the best of our knowledge, this is the first attempt to perform subspace clustering with irrelevant features.

## 2 Problem setup and method

### 2.1 Notations and model

The clean data matrix is denoted by $X_A \in R^{D \times N}$, where each column corresponds to a data point, normalized to a unit vector. The data points are lying on a union of $L$ subspace $S = \cup_{l=1}^{L} S_l$. Each subspace $S_l$ is of dimension $d_l$ which is smaller than $D$ and contains $N_l$ data samples with $N_1 + N_2 + \cdots + N_L = N$. We denote the observed dirty data matrix by $X \in R^{(D+D_1) \times N}$. Out of the $D + D_1$ features, up to $D_1$ of them are irrelevant. Without loss of generality, let $X = [X_O^T, X_A^T]^T$, where $X_O \in R^{D_1 \times N}$ denotes the irrelevant data. The subscript $A$ and $O$ denote the set of row indices corresponding to true and irrelevant features and the superscript T denotes the transpose. Notice that we do not know $O$ *a priori* except its cardinality is $D_1$. The model is illustrated in Figure 1. Let $X_A^{(l)} \in R^{D \times N_l}$ denote the selection of columns in $X_A$ that belongs to $S_l$. Similarly, denote the corresponding columns in $X$ by $X^{(l)}$. Without loss of generality, let $X = [X^{(1)}, X^{(2)}, ..., X^{(L)}]$ be ordered. Further more, we use the subscript "$-i$"to describe a matrix that excludes the column $i$, e.g., $(X_A)_{-i}^{(l)} = [(x_A)_1^{(l)}, ..., (x_A)_{i-1}^{(l)}, (x_A)_{i+1}^{(l)}, ..., (x_A)_{N_l}^{(l)}]$. We use the superscript $l^c$ to describe a matrix that excludes column in subspace $l$, e.g., $(X_A)^{l^c} = [X_A^{(1)}, ..., X_A^{(l-1)}, X_A^{(l+1)}, ..., X_A^{(L)}]$. For a matrix $\Sigma$, we use $\Sigma_{s,\eta}$ to denote the submatrix with row indices in set $s$ and column indices in set $\eta$. For any matrix $Z$, $P(Z)$ denotes the symmetrized convex hull of its column, i.e., $P(Z) = conv(\pm z_1, \pm z_2, ...., \pm z_N)$. We define $P_{-i}^l := P((X_A)_{-i}^{(l)})$ for simplification, i.e., the symmetrized convex hull of clean data in subspace $l$ except data $i$. Finally we use $\| \cdot \|_2$ to denote the $l_2$ norm of a vector and $\| \cdot \|_\infty$ to denote infinity norm of a vector or a matrix. Caligraphic letters such as $\mathcal{X}, \mathcal{X}_l$ represent the set containing all columns of the corresponding clean data matrix.

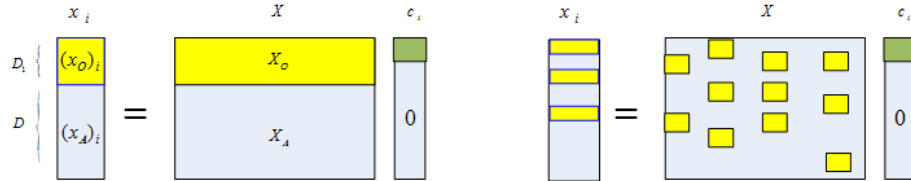

Figure 1: Illustration of the model of irrelevant features in the subspace clustering problem. The left one is the model addressed in this paper: Among total $D + D1$ features, up tp $D_1$ of them are irrelevant. The right one illustrates a more general case, where the value of any $D_1$ element of each column can be arbitrary (e.g., due to corruptions). It is a harder case and left for future work.

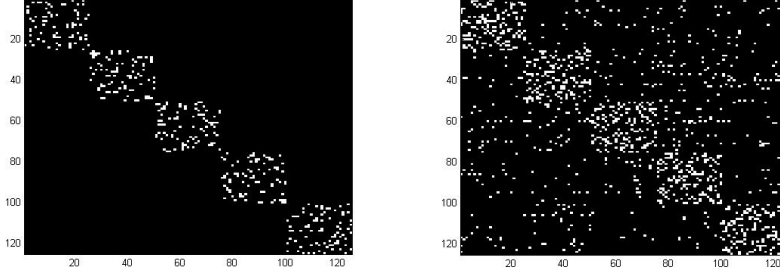

Figure 2: Illustration of the Subspace Detection Property. Here, each figure corresponds to a matrix where each column is $c_i$, and non-zero entries are in white. The left figure satisfies this property. The right one does not.

## 2.2 Method

In this secion we present our method as well as the intuition that derives it. When all observed data are clean, to solve the subspace clustering problem, the celebrated SSC [6] proposes to solve the following convex programming

$$\min_{c_i} \|c_i\|_1 \quad s.t. \quad x_i = X_{-i}c_i, \tag{1}$$

for each data point $x_i$. When data are corrupted by noise of small magnitude such as Gaussian noise, a straightforward extension of SSC is the Lasso type method called "Lasso-SSC" [18, 13]

$$\min_{c_i} \|c_i\|_1 + \frac{\lambda}{2}\|x_i - X_{-i}c_i\|_2^2. \tag{2}$$

Note that while Formulation (2) has the same form as Lasso, it is used to solve the subspace clustering task. In particular, the support recovery analysis of Lasso does not extend to this case, as $X_{-i}$ typically does not satisfy the RIP condition [18].

This paper considers the case where $X$ contains irrelevant/gross corrupted features. As we discussed above, Lasso is not robust to such corruption. An intuitive idea is to consider the following formulation first proposed for sparse linear regression [21].

$$\min_{c_i, E} \|c_i\|_1 + \frac{\lambda}{2}\|x_i - (X_{-i} - E)c_i\|_2^2 + \eta\|E\|_*, \tag{3}$$

where $\|\cdot\|_*$ is some norm corresponding to the sparse type of $E$. One major challenge of this formulation is that it is not convex. As such, it is not clear how to efficiently find the optimal solution, and how to analyze the property of the solution (typically done via convex analysis) in the subspace clustering task.

Our method is based on the idea of *robust inner product*. The robust inner product $\langle a, b \rangle_k$ is defined as follows: For vector $a \in R^D, b \in R^D$, we compute $q_i = a_i b_i, i = 1, ..., N$. Then $\{|q_i|\}$ are sorted and the smallest $(D - k)$ are selected. Let $\Omega$ be the set of selected indices, then $\langle a, b \rangle_k = \sum_{i \in \Omega} q_i$, i.e., the largest $k$ terms are truncated. Our main idea is to replace all inner products involved by robust counterparts $\langle a, b \rangle_{D_1}$, where $D_1$ is the upper bound of the number of irrelevant features. The intuition is that the irrelevant features with large magnitude may affect the correct subspace clustering. This simple truncation process will avoid this. We remark that we do not need to know the exact number of irrelevant feature, but instead only an upper bound of it.

Extending (2) using the robust inner product leads the following formulation:

$$\min_{c_i} \|c_i\|_1 + \frac{\lambda}{2}c_i^T \hat{\Sigma} c_i - \lambda \hat{\gamma}^T c_i, \tag{4}$$

where $\hat{\Sigma}$ and $\hat{\gamma}$ are robust counterparts of $X_{-i}^T X_{-i}$ and $X_{-i}^T x_i$. Unfortunately, $\hat{\Sigma}$ may not be a positive semidefinite matrix, thus (4) is not a convex program. Unlike the work [4][8] which studies

non-convexity in linear regression, the difficulty of non-convexity in the subspace clustering task appears to be hard to overcome.

Instead we turn to the Dantzig Selector, which is essentially a linear program (and hence no positive semidefiniteness is required):

$$\min_{c_i} \|c_i\|_1 + \lambda \|X_{-i}^T(X_{-i}c_i - x_i)\|_\infty. \tag{5}$$

Replace all inner product by its robust counterpart, we propose the following Robust Dantzig Selector, which can be easily recast as a linear program:

**Robust Dantzig Selector:** $$\min_{c_i} \|c_i\|_1 + \lambda \|\hat{\Sigma}c_i - \hat{\gamma}\|_\infty, \tag{6}$$

**Subspace Detection Property:** To measure whether the algorithm is successful, we define the criterion *Subspace Detection Property* following [18]. We say that the Subspace Detection Property holds, if and only if for all $i$, the optimal solution to the robust Dantzig Selector satisfies (1) **Non-triviality**: $c_i$ is not a zero vector; (2) **Self-Expressiveness Property**: nonzeros entries of $c_i$ correspond to only columns of $X$ sampled from the same subspace as $x_i$. See Figure 2 for illustrations.

## 3    Main Results

To avoid repetition and cluttered notations, we denote the following primal convex problem by $P(\Sigma, \gamma)$

$$\min_c \|c\|_1 + \lambda \|\Sigma c - \gamma\|_\infty.$$

Its dual problem, denoted by $D(\Sigma, \gamma)$, is

$$\max_\xi \langle \xi, \gamma \rangle \quad \text{subject to} \quad \|\xi\|_1 = \lambda \quad \|\Sigma \xi\|_\infty \leq 1. \tag{7}$$

Before we presents our results, we define some quantities.

The *dual direction* is an important geometric term introdcued in analyzing SSC [12]. Here we define similarly the dual direction of the robust Dantzig selector: Notice that the dual of robust Dantzig problem is $D(\hat{\Sigma}, \hat{\gamma})$, where $\hat{\gamma}$ and $\hat{\Sigma}$ are robust counterparts of $X_{-i}^T x_i$ and $X_{-i}^T X_{-i}$ respectively (recall that $X_{-i}$ and $x_i$ are the dirty data). We decompose $\hat{\Sigma}$ into two parts $\hat{\Sigma} = (X_A)_{-i}^T(X_A)_{-i} + \tilde{\Sigma}$, where the first term corresponds to the clean data, and the second term is due to the irrelevant features and truncation from the robust inner product. Thus, the second constraint of the dual problem becomes $\|((X_A)_{-i}^T(X_A)_{-i} + \tilde{\Sigma})\xi\|_\infty \leq 1$. Let $\xi$ be the optimal solution to the above optimization problem, we define $v(x_i, X_{-i}, \lambda) := (X_A)_{-i}\xi$ and the *dual direction* as $v^l = \frac{v(x_i^l, X_{-i}^{(l)}, \lambda)}{\|v(x_i^l, X_{-i}^{(l)}, \lambda)\|_2}$.

Similarly as SSC [12], we define the *subspace incoherence*. Let $V^l = [v_1^l, v_2^l, ..., v_{N_l}^l]$. The incoherence of a point set $\mathcal{X}_l$ to other clean data points is defined as $\mu(\mathcal{X}_l) = \max_{k:k \neq l} \|(X_A^{(k)})^T V^l\|_\infty$.

Recall that we decompose $\hat{\Sigma}$ and $\hat{\gamma}$ as $\hat{\Sigma} = (X_A)_{-i}^T(X_A)_{-i} + \tilde{\Sigma}$ and $\hat{\gamma} = (X_A)_{-i}^T(x_A)_i + \tilde{\gamma}$. Intuitively, for robust Dantzig selecter to succeed, we want $\tilde{\Sigma}$ and $\tilde{\gamma}$ not too large. Particularly, we assume $\|(x_A)_i\|_\infty \leq \epsilon_1$ and $\|(X_A)_{-i}\|_\infty \leq \epsilon_2$.

**Theorem 1** (Deterministic Model). *Denote* $\mu_l := \mu(\mathcal{X}_l)$, $r_l := \min_{i:x_i \in \mathcal{X}_l} r(P_{-i}^l)$, $r := \min_{l=1,...,L} r_l$ *and suppose* $\mu_l < r_l$ *for all l. If*

$$\frac{1}{r^2 - 4D_1\epsilon_1\epsilon_2 r - 2D_1\epsilon_2^2} < \min_l \frac{r_l - u_l}{2D_1\epsilon_2^2(u_l + r_l)}, \tag{8}$$

*then the subspace detection property holds for all $\lambda$ in the range*

$$\frac{1}{r^2 - 4D_1\epsilon_1\epsilon_2 r - 2D_1\epsilon_2^2} < \lambda < \min_l \frac{r_l - u_l}{2D_1\epsilon_2^2(u_l + r_l)}. \tag{9}$$

In an ideal case when $D_1 = 0$, the condition of the upper bound of $\lambda$ reduces to $r_l > u_l$, similar to the condition for SSC in the noiseless case [12].

Based on Condition (8), under a randomized generative model, we can derive how many irrelevant features can be tolerated.

**Theorem 2** (Random model). *Suppose there are $L$ subspaces and for simplicity, all subspaces have same dimension $d$ and are chosen uniformly at random. For each subspace there are $\rho d + 1$ points chosen independently and uniformly at random. Up to $D_1$ features of data are irrelevant. Each data point (including true and irrelevant features) is independent from other data points. Then for some universal constants $C_1, C_2$, the subspace detection property holds with probability at least $1 - \frac{4}{N} - N \exp(-\sqrt{\rho}d)$ if*

$$d \leq \frac{Dc^2(\rho) \log(\rho)}{12 \log N},$$

*and*

$$\frac{1}{\frac{1}{2}c^2(\rho)\frac{\log \rho}{d} - (\sqrt{2}c(\rho)\sqrt{\frac{\log \rho}{d}} + 1)\frac{C_1 D_1(\log D + C_2 \log N)}{D}} < \lambda < \frac{1 - \kappa}{1 + \kappa}\frac{D}{C_1 D_1(\log D + C_2 \log N)},$$

*where $\kappa = \sqrt{\frac{12d \log N}{Dc^2(\rho) \log \rho}}$ ; $c(\rho)$ is a constant only depending on the density of data points on subspace and satisfies (1) $c(\rho) > 0$ for all $\rho > 1$, (2) there is a numerical value $\rho_0$, such that for all $\rho > \rho_0$, one can take $c(\rho) = 1/\sqrt{8}$.*

*Simplifying the above conditions, we can determine the number of irrelevant features that can be tolerated. In particular, if $d \geq 2c^2(\rho) \log \rho$ and we choose the $\lambda$ as*

$$\lambda = \frac{4d}{c^2(\rho) \log \rho},$$

*then the maximal number of irrelevant feature $D_1$ that can be torelated is*

$$D_1 = \min\{\frac{c(\rho)D \log \rho}{8C_1 d(\log(D) + C_2 \log N)}, \frac{1 - \kappa}{1 + \kappa}\frac{C_0 Dc^2(\rho) \log \rho}{C_1 d(\log(D) + C_2 \log N)}\},$$

*with probability at least $1 - \frac{4}{N} - N \exp(-\sqrt{\rho}d)$.*

*If $d \leq 2c^2(\rho) \log \rho$, and we choose the same $\lambda$, then the number of irrelevant feature we can tolerate is*

$$D_1 = \min\{\frac{Dc(\rho)\sqrt{\frac{\log \rho}{d}}}{4\sqrt{2}C_1(\log(D) + C_2 \log N)}, \frac{1 - \kappa}{1 + \kappa}\frac{C_0 Dc^2(\rho) \log \rho}{C_1 d(\log(D) + C_2 \log N)}\},$$

*with probability at least $1 - \frac{4}{N} - N \exp(-\sqrt{\rho}d)$.*

**Remark 1.** If $D$ is much larger than $D_1$, the lower bound of $\lambda$ is proportional to the subspace dimension $d$. When $d$ increases, the upper bound of $\lambda$ decreases, since $\frac{1-\kappa}{1+\kappa}$ decreases. Thus the valid range of $\lambda$ shrinks when $d$ increases.

**Remark 2.** Ignoring the logarithm terms, when $d$ is large, the tolerable $D_1$ is proportional to $\min(C_1 \frac{1-\kappa}{1+\kappa}\frac{D}{d}, C_2 \frac{D}{d})$. When $d$ is small, $D_1$ is proportional to $\min(C_1 \frac{1-\kappa}{1+\kappa}\frac{D}{d}, C_2 D/\sqrt{d})$ .

## 4  Roadmap of the Proof

In this section, we lay out the roadmap of proof. In specific we want to establish the condition with the number of irrelevant features, and the structure of data (i.e., the incoherence $\mu$ and inradius $r$) for the algorithm to succeed. Indeed, we provide a lower bound of $\lambda$ such that the optimal solution $c_i$ is not trivial; and an upper bound of $\lambda$ so that the Self-Expressiveness Property holds. Combining them together established the theorems.

## 4.1 Self-Expressiveness Property

The Self-Expressiveness Property is related to the upper bound of $\lambda$. The proof technique is inspired by [18] and [12], we first establish the following lemma, which provides a sufficient condition such that Self-Expressiveness Property holds of the problem 6.

**Lemma 1.** *Consider a matrix $\Sigma \in R^{N \times N}$ and $\gamma \in R^{N \times 1}$, If there exist a pair $(\tilde{c}, \xi)$ such that $\tilde{c}$ has a support $S \subseteq T$ and*

$$
\begin{aligned}
&sgn(\tilde{c}_s) + \Sigma_{s,\eta}\xi_\eta = 0, \\
&\|\Sigma_{s^c \cap T,\eta}\xi_\eta\|_\infty \le 1, \\
&\|\xi\|_1 = \lambda, \\
&\|\Sigma_{T^c,\eta}\xi_\eta\|_\infty < 1,
\end{aligned}
\tag{10}
$$

*where $\eta$ is the set of indices of entry $i$ such that $|(\Sigma\tilde{c} - \gamma)_i| = \|\Sigma\tilde{c} - \gamma\|_\infty$, then for all optimal solution $c^*$ to the problem $P(\Sigma, \gamma)$, we have $c^*_{T^c} = 0$.*

The variable $\xi$ in Lemma 1 is often termed the "dual certificate". We next consider an oracle problem $P(\hat{\Sigma}_{l,l}, \hat{\gamma}_l)$, and use its dual optimal variable denoted by $\hat{\xi}$, to construct such a dual certificate. This candidate satisfies all conditions in the Lemma 1 automatically except to show

$$
\|\hat{\Sigma}_{l^c,\hat{\eta}}\hat{\xi}_{\hat{\eta}}\|_\infty < 1,
\tag{11}
$$

where $l^c$ denotes the set of indices expect the ones corresponding to subspace $l$. We can compare this condition with the corresponding one in analyzing SSC, in which one need $\|(X)^{(l^c)T}v\|_\infty < 1$, where $v$ is the dual certificate. Recall that we can decompose $\hat{\Sigma}_{l^c,\hat{\eta}} = (X_A)^{(l^c)T}(X_A)_{\hat{\eta}} + \tilde{\Sigma}_{l^c,\hat{\eta}}$. Thus Condition 11 becomes

$$
\|(X_A)^{(l^c)T}((X_A)_{\hat{\eta}}\hat{\xi}_{\hat{\eta}}) + \tilde{\Sigma}_{l^c,\hat{\eta}}\hat{\xi}_{\hat{\eta}}\|_\infty < 1.
\tag{12}
$$

To show this holds, we need to bound two terms $\|(X_A)_{\hat{\eta}}\hat{\xi}_{\hat{\eta}}\|_2$ and $\|\tilde{\Sigma}_{l^c,\hat{\eta}}\hat{\xi}_{\hat{\eta}}\|_\infty$.

**Bounding $\|\tilde{\Sigma}\|_\infty, \|\tilde{\gamma}\|_\infty$**

The following lemma relates $D_1$ with $\|\tilde{\Sigma}\|_\infty$ and $\|\tilde{\gamma}\|_\infty$.

**Lemma 2.** *Suppose $\hat{\Sigma}$ and $\hat{\gamma}$ are robust counterparts of $X^T_{-i}X_{-i}$ and $X^T_{-i}x_i$ respectively and among $D + D_1$ features, up to $D_1$ are irrelevant. We can decompose $\hat{\Sigma}$ and $\hat{\gamma}$ into following form $\hat{\Sigma} = (X_A)^T_{-i}(X_A)_{-i} + \tilde{\Sigma}$ and $\hat{\gamma} = (X_A)^T_{-i}(x_A)_i + \tilde{\gamma}$. We define $\delta_1 := \|\tilde{\gamma}\|_\infty$ and $\delta_2 := \|\tilde{\Sigma}\|_\infty$ .If $\|(x_A)_i\|_\infty \le \epsilon_1$ and $\|(X_A)_{-i}\|_\infty \le \epsilon_2$, then $\delta_2 \le 2D_1\epsilon_2^2$, $\delta_1 \le 2D_1\epsilon_1\epsilon_2$.*

We then bound $\epsilon_1$ and $\epsilon_2$ in the random model using the upper bound of the spherical cap [2]. Indeed we have $\epsilon_1 \le C_1(\log D + C_2 \log N)/\sqrt{D}$ and $\epsilon_2 \le C_1(\log D + C_2 \log N)/\sqrt{D}$ with high probability.

**Bounding $\|X_{\hat{\eta}}\hat{\xi}_{\hat{\eta}}\|_2$**

By exploiting the feasible condition in the dual of the oracle problem, we obtain the following bound:

$$
\|X_{\hat{\eta}}\hat{\xi}_{\hat{\eta}}\|_2 \le \frac{1 + 2D_1\lambda\epsilon_2^2}{r(P^l_{-i})}.
$$

Furthermore, $r(P^l_{-i})$ can be lower bound by $\frac{c(\rho)}{\sqrt{2}}\sqrt{\frac{\log \rho}{d}}$ and $\epsilon_2$ can be upper bounded by $C_1(\log D + C_2 \log N)/\sqrt{D}$ in the random model with high probability. Thus the RHS can be upper bounded.

Plugging this upper bound into (12), we obtain the upper bound of $\lambda$.

## 4.2 Non-triviality with sufficiently large $\lambda$

To ensure that the solution is not trivial (i.e., not all-zero), we need a lower bound on $\lambda$.

If $\lambda$ satisfies the following condition, the optimal solution to problem 6 can not be zero

$$\lambda > \frac{1}{r^2(P^l_{-i}) - 2D_1\epsilon_2^2 - 4r(P^l_{-i})D_1\epsilon_1\epsilon_2}. \tag{13}$$

The proof idea is to show when $\lambda$ is large enough, the trivial solution $c = 0$ can not be optimal. In particular, if $c = 0$, the corresponding value in the primal problem is $\lambda\|\hat{\gamma}_l\|_\infty$. We then establish a lower bound of $\|\hat{\gamma}_l\|_\infty$ and a upper bound of $\|c\|_1 + \lambda\|\hat{\Sigma}_{l,l}c - \hat{\gamma}_l\|_\infty$ so that the following inequality always holds by some carefully choosen $c$.

$$\|c\|_1 + \lambda\|\hat{\Sigma}_{l,l}c - \hat{\gamma}_l\|_\infty < \lambda\|\hat{\gamma}_l\|_\infty. \tag{14}$$

We then further lower bound the RHS of Equation (13) using the bound of $\epsilon_1$, $\epsilon_2$ and $r(P^l_{-i})$. Notice that condition (14) requires that $\lambda > A$ and condition (11) requires $\lambda < B$, where $A$ and $B$ are some terms depending on the number of irrelevant features. Thus we require $A < B$ to get the maximal number of irrelevant features that can be tolerated.

## 5 Numerical simulations

In this section, we use three numerical experiments to demonstrate the effectiveness of our method to handle irrelevant/corrupted features. In particular, we test the performance of our method and effect of number of irrelevant features and dimension subspaces $d$ with respect to different $\lambda$. In all experiments, the ambient dimension $D = 200$, sample density $\rho = 5$, the subspace are drawn uniformly at random. Each subspace has $\rho d + 1$ points chosen independently and uniformly random. We measure the success of the algorithms using the relative violation of the subspace detection property defined as follows,

$$RelViolation(C, \mathcal{M}) = \frac{\sum_{(i,j)\notin\mathcal{M}} |C|_{i,j}}{\sum_{(i,j)\in\mathcal{M}} |C|_{i,j}},$$

where $C = [c_1, c_2, ..., c_N]$, $\mathcal{M}$ is the ground truth mask containing all $(i, j)$ such that $x_i, x_j$ belong to a same subspace. If $RelViolation(C, \mathcal{M}) = 0$, then the subspace detection property is satisfied. We also check whether we obtain a trivial solution, i.e., if any column in $C$ is all-zero.

We first compare the robust Dantzig selector($\lambda = 2$) with SSC and LASSO-SSC ( $\lambda = 10$). The results are shown in Figure 3. The X-axis is the number of irrelevant features and the Y-axis is the Relviolation defined above. The ambient dimension $D = 200$, $L = 3$, $d = 5$, the relative sample density $\rho = 5$. The values of irrelevant features are independently sampled from a uniform distribution in the region $[-2.5, 2.5]$ in (a) and $[-10, 10]$ in (b). We observe from Figure 3 that both SSC and Lasso SSC are very sensitive to irrelevant information. (Notice that RelViolation=0.1 is pretty large and can be considered as clustering failure.) Compared with that, the proposed Robust Dantzig Selector performs very well. Even when $D_1 = 20$, it still detects the true subspaces perfectly. In the same setting, we do some further experiments, our method breaks when $D_1$ is about 40. We also do further experiment for Lasso-SSC with different $\lambda$ in the supplementary material to show Lasso-SSC is not robust to irrelevant features.

We also examine the relation of $\lambda$ to the performance of the algorithm. In Figure 4a, we test the subspace detection property with different $\lambda$ and $D_1$. When $\lambda$ is too small, the algorithm gives a trivial solution (the black region in the figure). As we increase the value of $\lambda$, the corresponding solutions satisfy the subspace detection property (represented as the white region in the figure). When $\lambda$ is larger than certain upper bound, $RelViolation$ becomes non-zero, indicating errors in subspace clustering. In Figure 4b, we test the subspace detection property with different $\lambda$ and $d$. Notice we rescale $\lambda$ with $d$, since by Theorem 3, $\lambda$ should be proportional to $d$. We observe that the valid region of $\lambda$ shrinks with increasing $d$ which matches our theorem.

## 6 Conclusion and future work

We studied subspace clustering with irrelevant features, and proposed the "robust Dantzig selector" based on the idea of robust inner product, essentially a truncated version of inner product to avoid

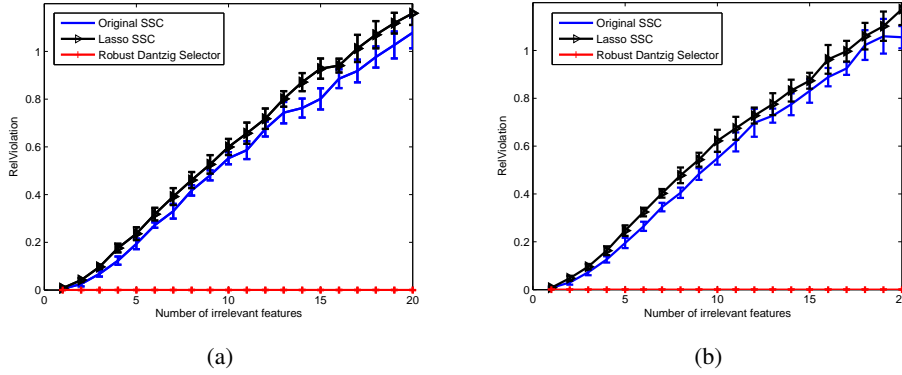

|     |     |
| :-: | :-: |
| (a) | (b) |

Figure 3: Relviolation with different $D_1$. Simulated with $D = 200$, $d = 5$, $L = 3$, $\rho = 5$, $\lambda = 2$, and $D_1$ from 1 to 20.

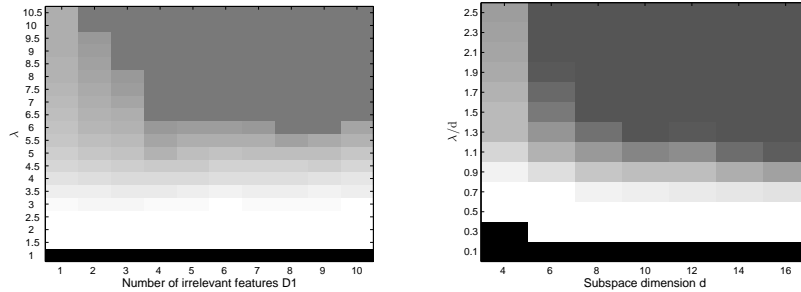

(a) Exact recovery with different number of irrelevant features. Simulated with $D = 200$, $d = 5$, $L = 3$, $\rho = 5$ with an increasing $D1$ from 1 to 10. **Black region**: trivial solution. **White region**: Non-trivial solution with RelViolation=0. **Gray region**: RelViolation> 0.02.

(b) Exact recovery with different subspace dimension d. Simulated with $D = 200$, $L = 3$, $\rho = 5$, $D_1 = 5$ and an increasing $d$ from 4 to 16. **Black region**: trivial solution. **White region**: Non-trivial solution with RelViolation=0. **Gray region**: RelViolation> 0.02.

Figure 4: Subspace detection property with different $\lambda$, $D_1$, $d$.

any single entry having too large influnce on the result. We established the sufficient conditions for the algorithm to exactly detect the true subspace under the deterministic model and the random model. Simulation results demonstrate that the proposed method is robust to irrelevant information whereas the performance of original SSC and LASSO-SSC significantly deteriorates.

We now outline some directions of future research. An immediate future work is to study theoretical guarantees of the proposed method under the semi-random model, where each subspace is chosen deterministically, while samples are randomly distributed on the respective subspace. The challenge here is to bound the subspace incoherence, previous methods uses the rotation invariance of the data, which is not possible in our case as the robust inner product is invariant to rotations.

### Acknowledgments

This work is partially supported by the Ministry of Education of Singapore AcRF Tier Two grant R-265-000-443-112, and A*STAR SERC PSF grant R-265-000-540-305.

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
