[Supplementary Material]

# Supplementary Material of Subspace Clustering with Irrelevant Features via Robust Dantzig Selector

**Chao Qu**
Department of Mechanical Engineering
National University of Singapore

A0117143@u.nus.edu

**Huan Xu**
Department of Mechanical Engineering
National University of Singapore

mpexuh@nus.edu.sg

Table 1: Table of Symbol and Notations

| | |
|---|---|
| $S_l$ | Subspace $l$ |
| $X_A$ | Clean data matrix |
| $X_O$ | Irrelevant data |
| $(X_A)_{-i}$ | A submatrix of $X_A$ that excludes the column $i$. |
| $X_A^l$ | The selection of columns in $X_A$ that belongs to $S_l$. |
| $\hat{\Sigma}$ | Robust counterpart of $X_{-i}^T X_{-i}$ |
| $\hat{\gamma}$ | Robust counterpart of $X_{-i}^T x_i$ |
| $P_{-i}^l$ | Symmetrized convex hull of $(X_A)_{-i}^{(l)}$ |
| $\Sigma_{s,\eta}$ | The submatrix of $\Sigma$ with row indices in set $s$ and column indices in set $\eta$. |
| $(X_A)^{l^c}$ | The submatrix matrix of $X_A$ that excludes column in subspace $l$. |
| $C_1, C_2, C_3$ | Numerical constants |
| $\|\cdot\|_\infty$ | Infinity norm of a vector or matrix |
| $\|\cdot\|_2$ | 2-norm of a vector |
| $\|\cdot\|_1$ | 1-norm of a vector |

**Lemma 1.** *Consider a matrix $\Sigma \in R^{N \times N}$ and $\gamma \in R^{N \times 1}$, If there exist a pair $(\tilde{c}, \xi)$ and $\tilde{c}$ has a support $S \subseteq T$, such that*

$$
\begin{aligned}
sgn(\tilde{c}_s) + \Sigma_{s,\eta}\xi_\eta &= 0, \\
\|\Sigma_{s^c \cap T, \eta}\xi_\eta\|_\infty &\leq 1, \\
\|\xi\|_1 &= \lambda, \\
\|\Sigma_{T^c,\eta}\xi_\eta\|_\infty &< 1,
\end{aligned}
\tag{1}
$$

*where $\eta$ is the set of indices of entry $i$ such that $|(\Sigma\tilde{c} - \gamma)_i| = \|\Sigma\tilde{c} - \gamma\|_\infty$, then for all optimal solution $c^*$ to the problem $P(\Sigma, \gamma)$, we have $c_{T^c}^* = 0$.*

*Proof.*

$$
\begin{aligned}
&\|c^*\|_1 + \lambda\|\Sigma c^* - \gamma\|_\infty - \|\tilde{c}\|_1 - \lambda\|\Sigma\tilde{c} - \gamma\|_\infty \\
&\geq \langle sgn(\tilde{c}_s), c_s^* - \tilde{c}_s \rangle + \|c_{s^c \cap T}^*\|_1 + \|c_{T^c}^*\|_1 + \langle \Sigma_{.,\eta}\xi_\eta, c^* - \tilde{c} \rangle \\
&= \langle -\Sigma_{s,\eta}\xi_\eta, c_s^* - \tilde{c}_s \rangle + \|c_{s^c \cap T}^*\|_1 + \|c_{T^c}^*\|_1 + \langle \Sigma_{.,\eta}\xi_\eta, c^* - \tilde{c} \rangle \\
&= \|c_{s^c \cap T}^*\|_1 + \|c_{T^c}^*\|_1 + \langle \Sigma_{s^c \cap T, \eta}\xi_\eta, c_{s^c \cap T}^* \rangle + \langle \Sigma_{T^c, \eta}\xi_\eta, c_{T^c}^* \rangle \\
&\geq (1 - \|\Sigma_{s^c \cap T, \eta}\xi_\eta\|_\infty)\|c_{s^c \cap T}^*\|_1 + (1 - \|\Sigma_{T^c, \eta}\xi_\eta\|_\infty)\|c_{T^c}^*\|_1. \\
&> 0(\text{unless}\|c_{T^c}^*\| = 0)
\end{aligned}
\tag{2}
$$

where the notion $\Sigma_{\cdot,\eta}$ denotes the submatrix of $\Sigma$ with same row and column indices in $\eta$. The first inequality holds from the convexity of objective function. The second inequality holds form the property of dual norm. The last inequality holds from the fact that $\|\Sigma_{T^c,\eta}\xi_\eta\|_\infty < 1$. Thus $c^*$ is not the optimal solution unless $c^*_{T^c} = 0$. Note that $\tilde{c}$ is also the optimal solution. $\square$

Before we start to prove our main theorem of the deterministic model, we first look at some properties of the robust inner product and how to decompose $\hat{\Sigma}$ and $\hat{\gamma}$.

**Lemma 2.** *Suppose $\hat{\Sigma}$ and $\hat{\gamma}$ are robust counterparts of $X^T_{-i}X_{-i}$ and $X^T_{-i}x_i$ respectively and among $D + D_1$ features, up to $D_1$ are irrelevant. We can decompose $\hat{\Sigma}$ and $\hat{\gamma}$ into following form $\hat{\Sigma} = (X_A)^T_{-i}(X_A)_{-i} + \tilde{\Sigma}$ and $\hat{\gamma} = (X_A)^T_{-i}(x_A)_i + \tilde{\gamma}$. We define $\delta_1 := \|\tilde{\gamma}\|_\infty$ and $\delta_2 := \|\tilde{\Sigma}\|_\infty$ .If $\|(x_A)_i\|_\infty \le \epsilon_1$ and $\|(X_A)_{-i}\|_\infty \le \epsilon_2$, then $\delta_2 \le 2D_1\epsilon_2^2$, $\delta_1 \le 2D_1\epsilon_1\epsilon_2$.*

*Proof.* Consider the robust inner product $h(j)$ between $jth$ column of $X_{-i}$ and $x_i$. $A$ is the set of indices of k such that $(X_{-i})_{kj}$ and $(x_i)_k$ are true data.

$$h(j) = \sum_{k \in A}(X_{-i})_{kj}(x_i)_k - \sum_{k \in \text{truncated true features}}(X_{-i})_{kj}(x_i)_k + \sum_{k \in \text{remaining irrelevant features}}(X_{-i})_{kj}(x_i)_k$$

(3)

Notice the first term is the inner product of the true data and we need to bound the last two terms.

For the wrongly truncated true data we have

$$\left| \sum_{k \in \text{truncated true features}}(X_{-i})_{kj}(x_i)_k \right| \le D_1(\max_{k \in A}|(X_{-i})_{kj}|)(\max_{k \in A}|(x_i)_k|).$$

A easy observation is

$$\left| \sum_{k \in \text{remaining irrelevant features}}(X_{-i})_{kj}(x_i)_k \right| \le \sum_{k \in \text{remaining irrelevant features}}|(X_{-i})_{kj}(x_i)_k|$$
$$\le \sum_{k \in \text{truncated true features}}|(X_{-i})_{kj}(x_i)_k| \quad (4)$$
$$\le D_1(\max_{k \in A}|(X_{-i})_{kj}|)(\max_{k \in A}|(x_i)_k|),$$

where the second inequality holds from the definition of the robust inner product.

So we can decompose $\hat{\gamma}$ into two parts

$$\hat{\gamma} = (X_A)^T_{-i}(x_A)_i + \tilde{\gamma},$$

where $\tilde{\gamma}_j = -\sum_{k \in \text{truncated true features}}(X_{-i})_{kj}(x_i)_k + \sum_{k \in \text{remaining irrelevant features}}(X_{-i})_{kj}(x_i)_k$. Thus

$$|\tilde{\gamma}_j| \le 2D_1(\max_{k \in A}|(X_{-i})_{kj}|)(\max_{k \in A}|(x_i)_k|) \le 2D_1\epsilon_1\epsilon_2$$

.

Similarly, we can decompose $\hat{\Sigma} = (X_A)^T_{-i}(X_A)_{-i} + \tilde{\Sigma}$. Consider the Robust inner project $h(p,q)$ between $pth$ column and $qth$ column of $X$. $A$ is the set of index $k$ such that $X_{kp}$ and $X_{kq}$ are true data.

$$h(p,q) = \sum_{k \in A}(X_{-i})_{kp}(X_{-i})_{kq} - \sum_{k \in \text{truncated true features}}(X_{-i})_{kp}(X_{-i})_{kq}$$
$$+ \sum_{k \in \text{remaining irrelevant features}}(X_{-i})_{kp}(X_{-i})_{kq}$$

(5)

The first term corresponds to the true data.

We define

$$\tilde{\Sigma}_{p,q} = - \sum_{k\in\text{truncated true features}} (X_{-i})_{kp}(X_{-i})_{kq} + \sum_{k\in\text{remaining irrelevant features}} (X_{-i})_{kp}(X_{-i})_{kq}.$$

Similarly we can bound last two terms.

$$\left| \sum_{k\in\text{truncated true features}} (X_{-i})_{kp}(X_{-i})_{kq} \right| \le D_1(\max_{k\in A}|(X_{-i})_{kp}|)(\max_{k\in A}|(X_{-i})_{kq}|).$$

$$\left| \sum_{k\in\text{remaining irrelevant features}} (X_{-i})_{kp}(X_{-i})_{kq} \right| \le \sum_{k\in\text{truncated true features}} |(X_{-i})_{kp}(X_{-i})_{kq}|$$
$$\le D_1(\max_{k\in A}|(X_{-i})_{kp}|)(\max_{k\in A}|(X_{-i})_{kq}|).$$

(6)

So we have

$$|\tilde{\Sigma}_{p,q}| \le 2D_1(\max_{k\in A}|(X_{-i})_{kp}|)(\max_{k\in A}|(X_{-i})_{kq}|) \le 2D_1\epsilon_2^2.$$

$\square$

It makes sense that upper bounded of $|\tilde{\Sigma}_{p,q}|$ and $|\tilde{\gamma}_j|$ are proportional to $D_1$ and $\|X_A\|_\infty$, since $D_1$ is the upper bound of the number of irrelevant features. It is easier to detect subspace with smaller $D_1$. $\|X_A\|_\infty$ decide the incoherence of data, small $\|X_A\|_\infty$ indicates the information spreads out. Suppose the true data matrix is sparse, then it is hard to know which feature is irrelevant and which one is the true.

# 1 Proof of Theorem 1

## 1.1 Construction of Dual Certificate

We consider the following oracle problem $P(\hat{\Sigma}_{l,l}, \hat{\gamma}_l)$ to construct the dual certificate.

$$\min_c \|c\|_1 + \lambda\|\hat{\Sigma}_{l,l}c - \hat{\gamma}_l\|_\infty, \tag{7}$$

where $\hat{\Sigma}_{l,l}$ and $\hat{\gamma}_l$ are robust counterparts of $(X_A)_{-i}^{(l)T}(X_A)_{-i}^{(l)}$ and $(X_A)_{-i}^{(l)T}(x_A)_i$ respectively.

The dual problem is

$$\max_\xi \langle \xi, \hat{\gamma} \rangle \quad \text{subject to} \quad \|\xi\|_1 = \lambda \quad \|\hat{\Sigma}_{l,l}\xi\|_\infty \le 1. \tag{8}$$

Optimal solution pair $(\hat{c}, \hat{\xi})$ of the this oracle problem satisfies

$$sgn(\hat{c}_{\hat{s}}) + \hat{\Sigma}_{\hat{s},\hat{\eta}}\hat{\xi}_{\hat{\eta}} = 0,$$
$$\|\hat{\Sigma}_{\hat{s}^c,\hat{\eta}}\hat{\xi}_{\hat{\eta}}\|_\infty \le 1, \tag{9}$$
$$\|\hat{\xi}\|_1 = \lambda,$$

where subscripts $\hat{s}$ and $\hat{\eta}$ denote the support of $\hat{c}$ and $\hat{\xi}$ respectively, and $\hat{\eta}$ is the set of indices of entry i such that $|(\Sigma_{l,l}\hat{c} - \hat{\gamma}_l)_i| = \|\Sigma_{l,l}\hat{c} - \hat{\gamma}_l\|_\infty$. If we set $\tilde{c} = (0, ..., \hat{c}, ..., 0)$ and $\xi = (0, ..., \hat{\xi}, ..., 0)$, it is easy to see we just need to verify the following condition for Lemma1.

$$\|\hat{\Sigma}_{l^c,\hat{\eta}}\hat{\xi}_{\hat{\eta}}\|_\infty < 1. \tag{10}$$

Recall that $\hat{\Sigma}_{l^c,\hat{\eta}} = (X_A)^{(l^c)T}(X_A)_{\hat{\eta}} + \tilde{\Sigma}_{l^c,\hat{\eta}}$. We define $v := (X_A)_{\hat{\eta}}\hat{\xi}_{\hat{\eta}}$. Thus the condition (10) becomes

$$\|(X_A)^{(l^c)T}v + \tilde{\Sigma}_{l^c,\hat{\eta}}\hat{\xi}_{\hat{\eta}}\|_\infty < 1. \tag{11}$$

we establish the condition required for (11) to hold. The idea is to provide a upper bound of the left hand side.

$$
\begin{aligned}
\|(X_A)^{(l^c)T}v + \tilde{\Sigma}_{l^c,\hat{\eta}}\hat{\xi}_{\hat{\eta}}\|_\infty &\leq \|(X_A)^{(l^c)T}v\|_\infty + \|\tilde{\Sigma}_{l^c,\hat{\eta}}\hat{\xi}_{\hat{\eta}}\|_\infty \\
&\leq \|v\|_2\|(X_A)^{(l^c)T}\frac{v}{\|v\|_2}\|_\infty + \|\tilde{\Sigma}_{l^c,\hat{\eta}}\|_\infty\|\hat{\xi}_{\hat{\eta}}\|_1 \\
&= \|v\|_2\|(X_A)^{(l^c)T}\frac{v}{\|v\|_2}\|_\infty + \lambda\|\tilde{\Sigma}_{l^c,\hat{\eta}}\|_\infty.
\end{aligned}
\tag{12}
$$

The term $\|(X_A)^{(l^c)T}\frac{v}{\|v\|_2}\|_\infty$ is the incoherence that we defined before. Now we need to bound $\|v\|_2$.

**Bounding $\|v\|_2$**

Before we bound $\|v\|_2$, we introduce the definition of polar set and circumradius.

**Definition of Polar Set**

The polar set $\mathcal{K}^\circ$ of the set $\mathcal{K} \in \mathcal{R}^D$ is defined as

$$
\mathcal{K}^\circ = \{y \in R^D : \langle x, y \rangle \leq 1 \text{ for all } x \in \mathcal{K}\}.
$$

**Definition of circumradius**

The circumradius of a convex body $P$, denoted by $\mathcal{R}(P)$, is defined as the radius of the smallest ball containing $P$.

We exploit the optimal condition in 9 to bound $\|v\|_2$. Using the first two condition in 9, we know

$$
\|\hat{\Sigma}_{l,\hat{\eta}}\hat{\xi}_{\hat{\eta}}\|_\infty \leq 1
$$

which implies

$$
\|(X_A)_{-i}^{(l)T}v + \tilde{\Sigma}_{l,\hat{\eta}}\hat{\xi}_{\hat{\eta}}\|_\infty \leq 1.
$$

The above condition implies

$$
\|(X_A)_{-i}^{(l)T}v\|_\infty \leq 1 + \|\tilde{\Sigma}_{l,\hat{\eta}}\hat{\xi}_{\hat{\eta}}\|_\infty
$$

and

$$
v \in [P(\frac{(X_A)_{-i}^{(l)}}{1 + \|\tilde{\Sigma}_{l,\hat{\eta}}\hat{\xi}_{\hat{\eta}}\|_\infty})]^\circ
$$

Using the definition of circumradius, we have

$$
\|v\|_2 \leq R([P(\frac{(X_A)_{-i}^{(l)}}{1 + \|\tilde{\Sigma}_{l,\hat{\eta}}\hat{\xi}_{\hat{\eta}}\|_\infty})]^\circ)
$$

The following lemma extracted from [2] relates the circumradius to the inradius of its polar set.

**Lemma 3.** *For a symmetric convex body $P$, i.e., $P = -P$, the following relationship between the inradius of $P$ and circumradius of its polar $P^\circ$ holds*

$$
r(P)R(P^\circ) = 1.
$$

Remind that polar of a polar set is the tightest convex envelope of original set, i.e., $\mathcal{K} = (\mathcal{K}^\circ)^\circ$. Combining this property and Lemma 3, we have

$$
\|v\|_2 \leq \frac{1 + \|\tilde{\Sigma}_{l,\hat{\eta}}\hat{\xi}_{\hat{\eta}}\|_\infty}{r(P_{-i}^l)} \leq \frac{1 + \|\hat{\xi}_{\hat{\eta}}\|_1\|\tilde{\Sigma}_{l,\hat{\eta}}\|_\infty}{r(P_{-i}^l)} = \frac{1 + \lambda\|\tilde{\Sigma}_{l,\hat{\eta}}\|_\infty}{r(P_{-i}^l)}
$$

## 1.2 The upper bound of $\lambda$

Replace $\|v\|_2$ in (12) by its upper bound we have

$$\frac{1 + \lambda\|\tilde{\Sigma}_{l,\hat{\eta}}\|_\infty}{r(P^l_{-i})}\mu_l + \lambda\|\tilde{\Sigma}_{l^c,\hat{\eta}}\|_\infty \leq 1 \tag{13}$$

Remind we define $\delta_2 = \|\tilde{\Sigma}\|_\infty$ and it is easy to see $\|\tilde{\Sigma}_{l^c,\hat{\eta}}\|_\infty \leq \delta_2$, $\|\tilde{\Sigma}_{l,\hat{\eta}}\|_\infty \leq \delta_2$. Thus a sufficient condition of (13) is

$$\lambda < \frac{r(P^l_{-i}) - u_l}{\delta_2(u_l + r(P^l_{-i}))}$$

Replace $\delta_2$ by its upper bound in Lemma 2, we get the upper bound of $\lambda$ in the theorem. If $\lambda$ is small, we can tolerate more irrelevant features. However $\lambda$ can not be as small as 0, otherwise $c = 0$ is the trivial solution of the primal problem. We discuss how to choose $\lambda$ to avoid this case in the following section.

## 1.3 The lower bound $\lambda$

If $\lambda$ satisfies the following condition, the optimal solution of Robust Dantzig Selector can not be zero

$$\lambda > \frac{1}{r^2(P^l_{-i}) - 2D_1\epsilon_2^2 - 4r(P^l_{-i})D_1\epsilon_1\epsilon_2}.$$

If $c = 0$ is the optimal solution in the primal problem $P(\hat{\Sigma}_{l,l}, \hat{\gamma}_l)$, the optimal value is $\lambda\|\hat{\gamma}_l\|_\infty$. Now we choose a special non-zero $c$ such that

$$\|c\|_1 + \lambda\|\hat{\Sigma}_{l,l}c - \hat{\gamma}_l\|_\infty < \lambda\|\hat{\gamma}_l\|_\infty. \tag{14}$$

Thus we can prove by contradiction. In particular, $c$ is the optimal solution of the following problem with clean data

$$\min_c \|c\|_1 \, s.t. \quad (X_A)^{(l)}_{-i}c = (x_A)_{-i}. \tag{15}$$

The dual of this problem is

$$\max_q \langle q, (x_A)_i \rangle \quad s.t. \|(X_A)^{(l)}_{-i}q\|_\infty \leq 1 \tag{16}$$

Since it is linear programming, the strong duality holds. We have $\langle q, (x_A)_i \rangle \leq \|q\|_2$ using the fact that $\|(x_A)_i\|_2 = 1$. Now we apply Lemma 3 again and use the definition of circumradius, the optimal value $\|c\|_1$ of (15) satisfies

$$\|c\|_1 \leq \|q\|_2 \leq \frac{1}{r(P^l_{-i})}.$$

We first look at the upper bound of LHS of (14).

$$
\begin{aligned}
&\|c\|_1 + \lambda\|\hat{\Sigma}_{l,l}c - \hat{\gamma}_l\|_\infty \\
=&\|c\|_1 + \lambda\|(X_A)^{(l)}_{-i}((X_A)^{(l)}_{-i}c - (x_A)_i) + \tilde{\Sigma}_{l,l}c - \tilde{\gamma}_l\|_\infty \\
=&\|c\|_1 + \lambda\|\tilde{\Sigma}_{l,l}c - \tilde{\gamma}_l\|_\infty \\
\leq&\|c\|_1 + \lambda\|\tilde{\Sigma}_{l,l}c\|_\infty + \lambda\|\tilde{\gamma}_l\|_\infty \\
\leq&\|c\|_1 + \lambda\|\tilde{\Sigma}_{l,l}\|_\infty\|c\|_1 + \lambda\|\tilde{\gamma}_l\|_\infty \\
\leq&(1 + \lambda\|\tilde{\Sigma}_{l,l}\|_\infty)\frac{1}{r(P^l_{-i})} + \lambda\|\tilde{\gamma}_l\|_\infty,
\end{aligned}
\tag{17}
$$

where the second equality holds from the fact that $c$ is feasible in problem (15).

Now we derive the lower bound of RHS of (14).

$$
\begin{aligned}
\lambda\|\hat{\gamma}_l\|_\infty &= \lambda\|(X_A)_{-i}^{(l)T}(x_A)_i + \tilde{\gamma}\|_\infty \\
&\geq \lambda\|(X_A)_{-i}^{(l)T}(x_A)_i\|_\infty - \lambda\|\tilde{\gamma}\|_\infty \\
&\geq \lambda r(P_{-i}^l) - \lambda\|\tilde{\gamma}\|_\infty,
\end{aligned}
\tag{18}
$$

where the last inequality holds from the geometric meaning of the inradius of a symmetric convex body.

Thus a sufficient condition for 14 is

$$
(1 + \lambda\|\tilde{\Sigma}_{l,l}\|_\infty)\frac{1}{r(P_{-i}^l)} + \lambda\|\tilde{\gamma}_l\|_\infty \leq \lambda r(P_{-i}^l) - \lambda\|\tilde{\gamma}\|_\infty,
$$

which implies

$$
\lambda > \frac{1/r(P_{-i}^l)}{r(P_{-i}^l) - 2\|\tilde{\gamma}_l\|_\infty - \frac{1}{r(P_{-i}^l)}\|\tilde{\Sigma}_{l,l}\|_\infty}.
\tag{19}
$$

Remind that we define $\delta_1 = \|\tilde{\gamma}\|_\infty$ and $\|\tilde{\gamma}_l\|_\infty \leq \delta_1$, we get the lower bound of $\lambda$

$$
\lambda > \frac{1}{r^2(P_{-i}^l) - \delta_2 - 2r(P_{-i}^l)\delta_1}.
$$

replace $\delta_1$ and $\delta_2$ by corresponding upper bounds in Lemma 2, we get the theorem.

## 2 Proof of Theorem 2

In this section, we prove Theorem 2 in the fully random model. In this case, both the orientation of the subspace and the distribution of the points are random. Before the proof, we need the following Lemma of the upper bound on the spherical cap [1].

**Lemma 4.** *Let $x \in R^D$ be a random vector sampled from a unit sphere and z is a fixed vector. Then we have*

$$
Pr(|x^T z| \geq \epsilon\|z\|_2) \leq 2\exp(\frac{-D\epsilon^2}{2}).
$$

### 2.1 Bounding $\delta_1$ and $\delta_2$

Remind that

$$
|\tilde{\gamma}_j| \leq 2D_1(\max_{k\in A}|(X_{-i})_{kj}|)(\max_{k\in A}|(x_i)_k|)
$$
$$
|\tilde{\Sigma}_{p,q}| \leq 2D_1(\max_{k\in A}|(X_{-i})_{kp}|)(\max_{k\in A}|(X_{-i})_{kq}|)
$$

Using Lemma 4 with $z = e_i$ ($e_i$ is the vector with a 1 in the ith coordinate and 0's elsewhere) and union bound over dimension $D$, we have

$$
Pr(\max_i|(X_A)_{ij}| \geq \epsilon) \leq 2D\exp(\frac{-D\epsilon^2}{2})
$$

for a fixed $j$.

In particular, we choose $\epsilon = \sqrt{\frac{\log(D)+8\log N}{D}}$. Using the union bound again over all $N^2$ entires of $\tilde{\Sigma}$, then we have

$$
\delta_1 \leq \frac{C_1 D_1(\log(D) + C_2\log(N))}{D},
$$

$$\delta_2 \le \frac{C_1 D_1 (\log(D) + C_2 \log(N))}{D}.$$

with probability at least $1 - \frac{1}{N}$ respectively, where $C_1$ and $C_2$ are some absolute constants.

## 2.2 Bounding the inradius $r$

We need to bound the inradius $r$, the following lemma is extracted from [2].

**Lemma 5.** *For every $\rho > 0$, there exist a constant $c(\rho)$ such that if $(1 + \rho)d < N < d\exp(d/2)$, such that*

$$Pr\{r(P^l_{-i}) \le \frac{c(\rho)}{\sqrt{2}} \sqrt{\frac{\log \rho}{d}} \quad \text{for all pairs} \quad (l, i)\} \le \sum_{l=1}^{L} N_l \exp(-d^\beta N_l^{1-\beta}),$$

*where $\rho = \frac{N_l - 1}{d}$ is the relative number of iid samples. There is a numerical value $\rho_0$, such that for all $\rho > \rho_0$, one can take $c(\rho) = 1/\sqrt{8}$.*

This Lemma is from the Theorem 2.8 in [2]. We choose $\beta = \frac{1}{2}$. In the fully random model, we have $N_1 = N_2 = ... = N_l = \rho d + 1$.

## 2.3 Incoherence bound

Notice $v$ just depends on the data $X^{(l)}_{-i}$ and $x_i$ corresponding to the subspace $l$, thus $v$ and $X^{(l^c)}_A$ are independent. Using the lemma 4, we have

$$Pr(\|x_A \frac{v}{\|v\|_2}\|_2 \le \sqrt{\frac{6 \log N}{D}}) \le \frac{2}{N^3}$$

for all true data point $x_A \in \mathcal{X} \backslash \mathcal{X}^l$ .

## 2.4 The range of $\lambda$

To make the upper bound of $\lambda$ meaningful, we need $r_l > u_l$. Replace $r_l$ by its lower bound and $u_l$ by its upper bound we have

$$d \le \frac{D c^2(\rho) \log(\rho)}{12 \log N}$$

and

$$\frac{1}{\frac{1}{2} c^2(\rho) \frac{\log \rho}{d} - (\sqrt{2} c(\rho) \sqrt{\frac{\log \rho}{d}} + 1) \frac{C_1 D_1 (\log D + C_2 \log N)}{D}} < \lambda < \frac{1 - \kappa}{1 + \kappa} \frac{D}{C_1 D_1 (\log D + C_2 \log N)},$$

where $\kappa = \sqrt{\frac{12 d \log N}{D c^2(\rho) \log \rho}}$.

## 2.5 The number of irrelevant features

We further simplify the lower bound of $\lambda$.

If $D_1 < D$, $D_1 < \frac{c(\rho) D \log \rho}{8 C_1 d (\log(D) + C_2 \log N)}$ and $d \ge 2c(\rho)^2 \log \rho$, then we can choose

$$\lambda = \frac{4d}{c(\rho)^2 \log \rho}.$$

Remind the upper bound of $\lambda < \frac{r(P^l_{-i}) - u_l}{\delta_2 (u_l + r(P^l_{-i}))}$, we set

$$\frac{4d}{c(\rho)^2 \log \rho} = \frac{r(P_{-i}^l) - u_l}{\delta_2(u_l + r(P_{-i}^l))}.$$

Replace $r$ by its lower bound and $\delta_1, \delta_2$ by the corresponding upper bounds, using the union bound, we can get the number of irrelevant features we can tolerate is

$$D_1 = \frac{1 - \sqrt{\frac{12d \log N}{Dc^2(\rho) \log \rho}}}{1 + \sqrt{\frac{12d \log N}{Dc^2(\rho) \log \rho}}} \frac{C_0 Dc(\rho)^2 \log \rho}{C_1 d(\log(D) + C_2 \log N)}$$

with probability at least $1 - \frac{4}{N} - N \exp(-\sqrt{\rho}d)$.

Thus

$$D_1 = \min\{D, \frac{c(\rho)D \log \rho}{8C_1 d(\log(D) + C_2 \log N)}, \frac{1 - \sqrt{\frac{12d \log N}{Dc^2(\rho) \log \rho}}}{1 + \sqrt{\frac{12d \log N}{Dc^2(\rho) \log \rho}}} \frac{C_0 Dc(\rho)^2 \log \rho}{C_1 d(\log(D) + C_2 \log N)}$$

$$= \min\{\frac{C(\rho)D \log \rho}{8C_1 d(\log(D) + C_2 \log N)}, \frac{1 - \sqrt{\frac{12d \log N}{Dc^2(\rho) \log \rho}}}{1 + \sqrt{\frac{12d \log N}{Dc^2(\rho) \log \rho}}} \frac{C_0 Dc(\rho)^2 \log \rho}{C_1 d(\log(D) + C_2 \log N)}. \tag{20}$$

Similarly $D_1 < D$, $D_1 < \frac{Dc(\rho)\sqrt{\frac{\log \rho}{d}}}{4\sqrt{2}C_1(\log(D) + C_2 \log N)}$ and $d \leq 2c(\rho)^2 \log \rho$, then we choose

$$\lambda = \frac{4d}{c(\rho)^2 \log \rho}.$$

Using the union bound again, we have

$$D_1 = \min\{\frac{Dc(\rho)\sqrt{\frac{\log \rho}{d}}}{4\sqrt{2}C_1(\log(D) + C_2 \log N)}, \frac{1 - \sqrt{\frac{12d \log N}{Dc^2(\rho) \log \rho}}}{1 + \sqrt{\frac{12d \log N}{Dc^2(\rho) \log \rho}}} \frac{C_0 Dc(\rho)^2 \log \rho}{C_1 d(\log(D) + C_2 \log N)}\}.$$

with probability at least $1 - \frac{4}{N} - N \exp(-\sqrt{\rho}d)$.

Figure 1: We test LASSO-SSC with different lambda. Simulated with $D = 200$, $d = 5$, $L = 3$, $\rho = 5$, and $D_1$ from 1 to 20. Notice that RelViolation=0.1 is pretty large and can be considered as clustering failure.

## Additional Numerical Experiment

We test LASSO-SSC with different $\lambda$ in Figure 1 to demonstrate that LASSO-SSC is not robust to irrelevant features. The X-axis is the number of irrelevant features and the Y-axis is the Relviolation. The ambient dimension $D = 200$, $L = 3$, $d = 5$, the relative sample density $\rho = 5$. The values of irrelevant features are independently sampled from a uniform distribution in the region $[-10, 10]$.