[Reviews · NeurIPS 2015]

Submitted by Assigned_Reviewer_1

- Glad to see simple simulations illustrating that the SSC and Lasso SSC don't solve this kind of problem well - thanks.

- Surprised and a bit confused the lasso-SSC does worse than the original SSC in Figure 3. This appears to be for a fixed (that is, non-optimized) lambda=2 on the soft constraint for Lasso, that doesn't seem like a great comparison, I would expect a larger lambda to do better, but would have expected any non-zero lambda to do better than the original SSC. A comment on this would have been useful.

- It is unfortunate the method breaks around 20% irrelevant features in the simulations. An interesting case would be if there are many many (like 90% or 99%) irrelevant features.

- Generally well-written, but paper needs a careful comb: many minor typos.
Summary: Paper extends subspace clustering to handle irrelevant features using a robust inner product (drops k largest terms). They re-cast the resulting non-convex optimization problem as a linear program using the robust Dantzig selector.

Paper more theoretical than experimental, which is ok, but limits its awesome-ness.

Submitted by Assigned_Reviewer_2

The authors define a convex optimization problem for each data point and aim to recover the support of a data point in terms of other data points. The main contribution is to replace the standard inner product between two vectors with its robust counterpart, by selecting the smallest (hopefully) non-irrelevant features.

Comments - Could the authors please define 'robustness' in the abstract or briefly describe it.

- line 37 'motivates'

- line 77, lin 82, can club WLOG together, or show a picture.

- line 149, where does the robustness come from ? What is intuitively the benefit of neglecting the larger dot-products ?

- what happens when all irrelevant features are really small ? they would satisfy line 207 as well enter the dot-products, what would fail then ?

- The current set of numerical simulations although limited, show some insight. Could the authors also please the add the result of choosing the top 'D - k' for various values of 'k', rather than just D1 ? This would show the benefit of robustness in the first place.

Summary: This paper proposes to perform subspace clustering by formulating a convex optimization problem for each data point and using a 'robust' definition of inner product. Unfortunately, I am not very familiar with this area and am not able to judge the quality of this work very well.

Submitted by Assigned_Reviewer_3

The paper casts the subspace clustering problem as a Dantzig Selector estimator (along the line of Candes-Tao's work) and claims for its robustness w.r.t. irrelevant features. The paper is essentially theoretical, which provides guarantees for the algorithm to find the correct subspace. As I am not expert in the field of robust estimator for sparse recovery, it is difficult for me to evaluate the quality of this work.
Summary: The paper casts the subspace clustering problem as a Dantzig Selector estimator (along the line of Candes-Tao's work) and claims for its robustness w.r.t. irrelevant features.

Submitted by Assigned_Reviewer_4

The paper proposed to use robust Dantzig selector to perform subspace clustering with irrelevant features. The idea presented in the paper is straightforward and intuitive. The authors also provided analysis to show when the Robust Dantzig selector can detect the true subspace clustering. My concern on this work is the usefulness of the approach in real application. In the method, each sample is used as the target of the robust Dantzig selector. If the data is large, too many models need to be fit, which makes the methods impractical.
Summary: The paper proposed to use robust Dantzig selector to perform subspace clustering with irrelevant features. The idea presented in the paper is straightforward and intuitive, but when data size is large the proposed method might not be able to handle it.

Submitted by Assigned_Reviewer_5

For my understanding, there is no silver bullet for feature selection clustering algorithms: their performances are ultimately affected by the nature of datasets and tasks. Therefore it is very important for clustering works to show concrete examples where the proposed model works well, and where it does not work (limitations). In that sense, the paper does not validate the proposed model in real world datasets, and no such discussions are provided. This greatly reduces the reliability concerning the usefulness of the proposed model.

A number of irrelevant features D1 = 20 within D=200 is too "mild" to use for some applications such as sensor networks. In such a small setting, we may simply conduct "leave-one-feature-out" studies to identify irreverent features. It may make the experiments more convincing if you can work on more large D cases, or 99% of features are indeed irrelevant.
Summary: This paper proposes a variant of the sparse subspace clustering, to deal with the existence of the irrelevant feature attributes. The objective introduces a sparse error term into the SSC objective.

I think the proposed model is simple and reasonable. However, experimental validation is too weak to recommend for NIPS publication.

Author Feedback
Author rebuttal: We thank all reviewers for their detailed and constructive comments.

Reviewer 1 and 3 commented that it would be better if the algorithm can handle many (e.g., 90% or 99%) irrelevant features. We certainly agree with this. However, we want to point out that the subspace clustering problem with corrupted feature is an open problem, even when the fraction of irrelevant features are mild. Indeed, as shown in the paper, SSC and LASSO-SSC break down even with few irrelevant features. In this sense, our work opens a door to the area of subspace clustering with corrupted data, and hopefully may inspire methods that can eventually handle the much more challenging case of many irrelevant features, maybe under additional assumptions on the irrelevant features (recall that in this paper we made no assumptions on the irrelevant features).

We now respond to other comments:

Reviewer 1

Thank you for your suggestion on LASSO-SSC. We will fine-tune the parameter \lambda of LASSO-SSC and compare the results in the final version of the paper.

Reviewer 3

We admit that simulations on real world datasets would improve the paper. However, we want to point out that our work focuses on providing methods with theoretical guarantees (deterministic model and fully random model) to solve the subspace clustering problem with corrupted features. It is based upon the sparse subspace clustering, an algorithm that has been widely applied in many real problems.

As for your suggestion on "leave-one-feature-out " strategy, we think it is not easy to apply this to the subspace clustering problem with corrupted features. Even with D1=20, and D=200 (which is the setting in our simulation), if we follow "leave-one-feature-out " strategy, we need to enumerate C_{200}^{20} (i..e., approximately 10^46) combinations, and solve a Lasso-like algorithm for each. This is way beyond the existing computation power of the world. Moreover, scalability of such a strategy is clearly a problem.

Reviewer 4

Thank you for your comments. We now provide an intuitive explanation why we use this robust inner product. The essence of SSC, LASSO-SSC and our formulation is to write a sample as a sparse linear combination of other samples (plus some small noise) . Intuitively, the irrelevant features with large magnitude may affect the correct subspace clustering, so we introduce this truncation process of robust inner-product. When all irrelevant features are really small, they do enter the dot-products as you said. But since they are small, they would not affect the correct clustering, as they will be treated as "small noise" by the Lasso-like algorithm of Lasso-SSC. By exploiting this intuition, we bound the error terms \delta_1, \delta_2 in Lemma2, which only depends on the number of irrelevant features but not the magnitude, which lead to our theoretical guarantees.

Thank you for your suggestion on choosing top 'D-k' for various value of 'k' to illustrate robustness, we will add it in the final version of this paper.

Reviewer 6

Thank you for your comments. As for your concern about large computations in real application, we believe this should not be an issue. Notice that in terms of computation cost our method is similar to the original sparse subspace clustering (SSC) algorithm, which also needs to solve a linear programming for each sample. Yet, SSC has been widely used in computer vision and other real applications (where datasets are typically pretty large) and has seen many successes. Moreover, we can easily accelerate computations by parallel computing, since for each sample we solve an independent linear optimization problem.